# Lost in Translation: Evaluation of Subcutaneous Interferon-β Treatment for SARS-CoV-2 Infection in Real Life

**DOI:** 10.3390/jcm11236952

**Published:** 2022-11-25

**Authors:** José L. Casado, Pilar Vizcarra, José M. Del Rey, María Cruz Soriano, Mario Rodriguez-Dominguez, Luis Manzano, Julio Acero, Carmen Palomar-Fernandez, Alejandro Vallejo

**Affiliations:** 1Departamento de Enfermedades Infecciosas, Hospital Universitario Ramón y Cajal, IRYCIS, CIBERINFEC, 28034 Madrid, Spain; 2Departamento de Bioquímica, Hospital Universitario Ramón y Cajal, 28034 Madrid, Spain; 3Departamento de Medicina Intensiva, Hospital Universitario Ramón y Cajal, IRYCIS, 28034 Madrid, Spain; 4Departamento de Microbiología, Hospital Universitario Ramón y Cajal, IRYCIS, CIBERESP, 28034 Madrid, Spain; 5Departamento de Medicina Interna, Hospital Universitario Ramón y Cajal, IRYCIS, 28034 Madrid, Spain; 6Departamento de Cirugía Oral y Maxilo-Facial, Hospital Universitario Ramón y Cajal, IRYCIS, 28034 Madrid, Spain; 7Departamento de Farmacia, Hospital Universitario Ramón y Cajal, 28034 Madrid, Spain

**Keywords:** SARS-CoV-2, COVID-19, interferon-beta, real life, propensity score

## Abstract

Despite in vitro activity of interferon-β (IFN-β) against SARS-CoV-2 infection, its clinical efficacy remains controversial. We evaluated the impact of IFN-β treatment in a cohort of 3590 patients hospitalized with COVID-19 during March–April 2020. The primary endpoint was a composed variable of admission to intensive care unit (ICU)/death. Overall, 153 patients (4%) received IFN-β. They were significantly more severely ill, with a worse clinical and analytical situation, explaining a higher ICU admission (30% vs. 17%; *p <* 0.01), and a shorter time to the composed variable. In a Cox regression analysis, older age, lymphopenia, renal failure, or increased neutrophil-to-lymphocyte ratio were associated with a greater hazard ratio (HR) of admission at ICU/death. Notably, the HR of IFN-β for the outcome variable was no longer significant after adjustment (HR, 1.03; 95% CI, 0.82–1.30), and different sensitivity analysis (early IFN use, ICU admission) showed no changes in the estimates. A propensity score matching analysis showed no association of IFN-β therapy and outcome. In conclusion, in this large cohort of hospitalized COVID-19 patients, IFN-β was used mainly in patients with advanced disease, reflecting an important bias of selection. After adjusting by severity, IFN-β was not associated with a higher rate of ICU admission or mortality.

## 1. Introduction

In vitro studies have clearly shown that type 1 interferon (IFN) can inhibit SARS-CoV-2, and this fact explains why viral replication after infection leads to impaired induction of IFN [1]. Furthermore, a significant proportion of severely ill patients have neutralizing auto antibodies against these type I IFNs [2,3]. Thus, the use of IFN could be useful against SARS-CoV-2 infection, as observed in studies against other coronaviruses [4], but, since the beginning of the pandemic, in vitro data suggest that its use could be of risk in late stages of the disease because of the stimulation of proinflammatory signals [5,6]. Later studies have showed a benefit, neutral effects, or a worse evolution with the use of this drug [7,8,9]. Therefore, uncertainty about its clinical efficacy remains, limiting the use of a hypothetically effective drug that could be used as an antiviral and immunomodulator. To address the usefulness of a subcutaneous interferon-β, we evaluated its use and the results in a large cohort of COVID-19 patients attended to in a tertiary hospital.

## 2. Materials and Methods

We retrospectively evaluated the use of subcutaneous IFN-β-1b (0.25 mg, 8 million IU) every 48 h, in a cohort of 6481 COVID-19 attended patients in a tertiary university hospital during the first surge of the disease. A total of 2891 patients were excluded because of being not hospitalized, or had incomplete data about clinical presentation, analytical evolution, concomitant therapies, or outcome. Therefore, 3590 consecutive adult patients met the criteria of COVID-19 confirmed by real-time polymerase chain reaction (RT-PCR) assay, or high suspicion owing to clinical presentation, being hospitalized in our hospital during March and April 2020 and having following data until outcome after admission. All of the demographic, clinical, analytical, and microbiological data were collected after the outcome.

Study data were collected and managed using REDCap electronic data capture tools hosted at IRYCIS. REDCap (Research Electronic Data Capture) is a secure, web-based application designed to support data capture for research studies, providing (1) an intuitive interface for validated data entry; (2) audit trails for tracking data manipulation and export procedures; (3) automated export procedures for seamless data downloads to common statistical packages; and (4) procedures for importing data from external sources [10]. The study was approved by the Ethics Committee of our hospital with waived informed consent for the collection of clinical data.

The outcome variable was a composed variable of intensive care unit (ICU) admission or death during hospitalization and the time to the variable. This composed variable was chosen because of the differences in the ICU admission criteria during the first surge of the disease, thus trying to avoid the bias of considering a candidate to ICU or IFN use to those patients with better or worse probabilities of response. In any case, we separately analyze and compare the rate of, and time to, ICU admission and mortality during admission. The subcutaneous (sc) administration of IFN-β (interferon therapy, IT) was classified according to the moment of use as early IFN-β treatment (EIT) if started within <7 days since symptoms’ initiation, or late IFN-β treatment (LIT) if started from day 7 onwards to reflect an expected optimal time for using this drug, or no IFN-β treatment (no IT) if IFN-β was not provided. Additional exposure variables included demographic data, chronic underlying conditions, admission symptoms and signs, laboratory findings, concomitant treatment with corticosteroids, and severity.

### Statistical Analysis

The *χ*^2^ test or Fisher’s exact test were used to compare categorical variables. When appropriate, continuous variables were dichotomized. Cox regression was used to analyze the impact of IT on the composed variable ICU/death. Variables with a *p-*value < 0.10 in univariate comparisons and those considered of clinical importance were entered into the multivariate model. Interactions and collinearity were evaluated. Sensitivity analyses for ICU/mortality were performed including changes in covariables. In addition, a propensity score (PS) for receiving IT or no IFN therapy was calculated using the data of the multivariate analysis, and its predictive ability for observed data was assessed using the area under the receiver operating characteristic curve (AUROC) with a 95% confidence interval (CI). The PS was used to perform a matched cohort analysis in which patients undergoing IT and no interferon were paired (1:1) according to their PS using calipers of 0.01 standard deviation. The composed variable in the matched pairs was compared by Cox regression. All statistical analyses were carried out using SPSS software (SPSS 20.0, IBM, Armonk, NY, USA).

## 3. Results

Overall, 3590 hospitalized patients were included in the present study. Of them, 153 (4.3%) received IT, with a median of 8 days after symptoms’ initiation (IQR, 6–11) and 2 days after admission (0–10); 46 (30%%) received early IFN-β treatment (median [IQR] days from symptoms initiation 5 [3,4,5,6]), and 107 (70%) received late IFN-β treatment (median [IQR] days from symptoms initiation 10 [8,9,10,11]). In total, time on IFN therapy was 10 days (IQR, 4–17); therefore, hospitalized patients received up to five doses of interferon subcutaneously every other day. Characteristics of the patients receiving or not IFN are shown in Table 1.

Compared with patients receiving standard therapy, those starting IFN-β had a similar age, but were older, predominantly male, and had more comorbidities. Moreover, they were admitted with a significantly greater proportion of persistent fever; oxygen desaturation; and laboratory indicators of high risk such as renal involvement, lymphopenia, elevated levels of D-dimer, lactate dehydrogenase, C-reactive protein, and higher neutrophile-to-lymphocyte ratio (NLR) (*p <* 0.001 for all the variables). Thus, patients receiving IFN were prone to receive corticosteroids as concomitant therapy in a greater proportion. Of note, even patients receiving EIT continued to have a severe presentation, highlighting the differences between the different groups of therapy (Table 2).

As could be expected because of severity at the time of admission, more patients severely ill receiving IFN-β were admitted to ICU (30 vs. 17%; *p <* 0.001), without differences in time from hospitalization or symptoms to ICU admission, or severity as measured by SAPS-II or APACHE-II scores. In those receiving early INF use, despite a short evolution of the disease, the severe clinical and analytical presentation also led to ICU admission and higher mortality.

Moreover, overall mortality was 39% in those receiving IFN-β and 17% in the control group (*p <* 0.001); therefore, there was a significant difference in the composed outcome favoring those who had not received IFN-β (58 vs. 31%; *p* < 0.01. Of note, in the Cox multivariate analysis, the covariates significantly associated with a higher rate of ICU/mortality (Table 3) were older age (adjusted hazard ratio, aHR, 1.015 by additional year), eGFR < 60 mL/min/m^2^ (aHR 1.55), lymphopenia < 900 cells/mm (aHR 1.576), use of corticosteroids (aHR 0.929), C-reactive protein above 75 mg/L (aHR 1.694), and NLR (a1.012 by additional unit), whereas sex, days of symptoms, other significant analytical parameters, or low number of platelets were not associated. After adjustment in the Cox multivariate analysis, IFN-β treatment no longer showed an association with mortality (aHR, 1.149; 95% CI, 0.79–1.662; *p* = 0.461).

The estimations of the associations of EIT with mortality in sensitivity analyses were consistent with the analysis in the whole cohort. In different sensitivity analysis, including the variable IFN as early IFN-β use (HR 0.815; 95% CI, 0.316–2.01; *p =* 0.673), or even selecting only those with ICU admission, multivariate Cox proportional hazards models adjusting for these baseline characteristics did not produce substantially different estimates of the treatment effect, and the use of IFN-β was not associated with mortality (HR 0.853; *p =* 0.528).

Finally, we matched 42 pairs of patients receiving IT or not based on the propensity score. Matched patients had a similar exposure frequency to all variables (Table 4). Again, the use of IFN-β treatment was not associated with ICU/mortality in this analysis (aHR, 0.523; 95% CI, 0.217–1.26; *p* = 0.149).

## 4. Discussion

We tried to evaluate the role of subcutaneous IFN-β therapy in a large retrospective cohort of COVID-19 patients, in order to ascertain its effectiveness and safety in real life. We observed that, in our hospital, the use of IFN-β in the clinical setting was limited to advanced patients in the worse situation, where the possible efficacy of this drug seems to be clearly blunted. Thus, patients receiving IFN-β had more frequently severe symptoms and signs, high values of inflammatory biomarkers, and received respiratory and/or hemodynamic support in a higher proportion. Even with this limitation, IFN-β treatment did not show an association with mortality after adjustment with known variables such as age, or analytical parameters of severity.

As mentioned previously, data on the increased severity of COVID-19 in patients with no endogenous IFN-β and low IFN-α production or with neutralizing auto-Abs against these type I IFNs [3,11] have strongly suggested a potential role for IFN [12]. In line with this, randomized, well-designed studies showed a benefit for IFN use, such as statistically greater odds of improvement and significant reductions in time to clinical improvement [13,14], a significant increase in the rate of patients discharged by day 14 and a lower overall mortality at 28 days [7], or even how the combination group of IFN and lopinavir/r had a much faster clinical recovery and a narrower viral shedding window [15].

On the other hand, larger studies such as the Solidarity [16] and DisCoVeRy trials [17] did not find differences between interferon β-1a treatment and usual care and, recently, a randomized study showed no difference in mortality with the combination of remdesivir and IFN-β, and even a worse evolution in ICU patients [9], in line with the concerns over its pro-inflammatory side-effects if administered at later stages [6]. These differences were also suggested by a post-hoc analysis that indicated more substantial effects with an earlier initiation (less than seven days after the symptom onset of the combination therapy) [15]. Indeed, the different studies are not conclusive on the efficacy of interferon β-1a in patients with early-stage or mild disease, who do not require hospitalization, nor on the effect of interferon β-1a without concomitant use of steroids, and other routes of interferon administration, such as inhaled or intravenous, have been reported to have different mechanisms of action [18].

Although we used a robust primary outcome, ICU/mortality, to reflect both those patients’ candidates for intensive therapy and those with limitations to ICU owing to age or previous health conditions, most of our IFN-treated patients had ICU criteria at the time of admission. Thus, the efficacy of IFN-β could not be properly evaluated in this population. Even so, we found no differences in mortality after adjustment. In addition, we conducted a post hoc, propensity score-adjusted study of patients with COVID-19, investigating the effectiveness of subcutaneous INF-β treatment. Again, after adjustment, interferon therapy was not associated with mortality. Moreover, in a similar study including several centers in Spain, the use of IFN-β was not associated with mortality after adjusting by different variables, as well as after a propensity score matching was performed [8].

Our study showed the importance of a rapid evaluation of repurposing drugs in the context of a medical emergency. In our center, the initial in vitro negative results, as well as the subjective perception and hasty communications by some clinicians of a worse evolution for IFN-treated patients, lead to a restricted use of this drug, only for those having a more severe presentation. Thus, we confirm the potential for bias in treatment selection, patient assessment, and patient enrolment already observed in other open studies [16,17].

The present study has several limitations in addition to the selection bias discussed that limit our information about the efficacy of IFN in patients with early-stage or mild disease, or who do not require hospitalization. Moreover, our relatively small sample size of INF-treated patients precludes us from adequately adjusting the treatment effect by baseline differences. We divided early and late IFN use for identifying those with a possible better efficacy of this drug, as presented in similar studies, although this group of early-treated patients already presented a severe clinical and analytical situation at inclusion. Finally, we had no access to the follow-up RT-PCR; therefore, we were unable to determine the time to a negative test or shed further light on the effect of IFN-β in viral dynamics.

In conclusion, our findings showed the challenges in translating the in vitro benefit of subcutaneous IFN as therapy for COVID-19 to the clinical setting, at least in our milieu, because of the usually late use of this drug in real life. In any case, our data confirm that IFN-β was not associated with a worse evolution or mortality, even in patients admitted with a severe presentation of COVID-19. Our data could avoid the loss of this drug for the future and suggest the need for well-designed studies, such as the evaluation of specific subgroups in terms of IFN levels or clinical factors, for using drugs with complex mechanisms of action in new diseases such as COVID-19.

## Figures and Tables

**Table 1 jcm-11-06952-t001:** Features of hospitalized COVID-19 patients according to IFN use. Data are presented as median, interquartile range (IQR), No. (%). *p*-values are calculated by χ2, Fisher’s test, or Mann–Whitney’s U test.

Variable	IT	No IT	*p*-Value
	(*n* = 153)	(*n* = 3437)	(IT vs. No IT)
Female sex	49 (32)	1382 (40)	0.043
Age	69 (59–79)	68 (55–81)	0.697
BMI (Kg/m^2^)	29.3 (25.4–32.3)	27.9 (25.2–31.2)	0.04
Hypertension	28 (18)	412 (12)	0.021
Diabetes	18 (12)	275 (8)	0.09
Obesity (BMI > 30)	57 (42)	1169 (34)	0.2
Chronic kidney disease (eGFR < 60 mL/min/1.73 m^2^)	17 (11)	241 (7)	0.001
Days of symptoms until admission	6.5 (4–8)	7 (4–10)	0.442
Days from symptoms initiation to IFN	8 (6–11)	-	-
Fever at admission	112 (73)	1650 (48)	<0.001
Low SpO_2_ < 94% at admission	87 (57)	1375 (40)	<0.001
eGFR (MDRD)	73.2 (58–90.3)	78.5 (70–94.8)	0.122
eGFR < 60 mL/min/1.73 m^2^	44 (29)	859 (25)	0.07
Lymphocyte count < 900/μL	85 (56)	1547 (45)	<0.001
NLR	6.41 (3.3–13.2)	5.2 (2.9–9.9)	0.017
Platelets < 150,000/μL	34 (22)	653 (19)	<0.001
INR ≥ 1.3	28 (18)	3093 (9)	<0.001
D-dimer levels (ng/mL)	752 (481–1875)	640 (368–1272)	0.035
Lactate dehydrogenase ≥ 250 U/L	126 (82)	2509 (73)	0.003
C-reactive protein (mg/L)	278 (77–392)	70 (15–144)	<0.001
Procalcitonin	0.08 (0.01–1.3)	0.07 (0.01–0.8)	0.877
Corticosteroids	132 (86)	2084 (61)	<0.01
Time on CE therapy (days)	12 (6–20)	9 (5–19)	0.027
ICU	46 (30)	588 (17)	<0.001
Days from symptoms initiation to ICU	14 (9–22)	15 (10–21)	0.244
Days from admission to ICU	6 (1.5–13)	5 (2–10)	0.302
SAPS II at ICU admission	33.5 (27.5–43)	38 (30–49)	0.041
Apache II score at ICU admission	13 (10–18)	15 (11–21)	0.143
Mortality at day 30	60 (39)	574 (17)	<0.001
Composed outcome (ICU/Death)	88 (58)	1078 (31)	<0.001

Abbreviations: IT, interferon-β treatment; LIT; No IT, no interferon-β treatment; BMI, body mass index; eGFR, estimated glomerular filtration rate; SpO_2_, peripheral capillary oxygen saturation; NLR, neutrophil-to-lymphocyte ratio; INR, international normalized ratio; CE, corticosteroids; SAPS II, Simplified Acute Physiologic Score II; APACHE II, Acute Physiology and Chronic Health Disease Classification System II; ICU, intensive care unit.

**Table 2 jcm-11-06952-t002:** Characteristics of patients according to the use of IFN as early (<7 days of symptoms) or late IFN use. Data are presented as median, interquartile range (IQR), No. (%). *p*-values are calculated by *χ*^2^, Fisher’s test, or Mann–Whitney’s *U* test.

Variable	EIT (*n* = 46)	LIT (*n* = 107)	*p*-Value (EIT vs. LIT)
Female sex	16 (35)	33 (31)	0.632
Age	76 (60–83)	67 (58–77)	0.098
BMI (Kg/m^2^)	28.2 (25–30.3)	29.7 (25.7–32.8)	0.159
Hypertension	11 (24)	17 (16)	0.21
Diabetes	7 (16)	11 (10)	0.29
Obesity (BMI > 30)	14 (31)	43 (40)	0.123
Chronic kidney disease (eGFR < 60 mL/min/1.73 m^2^)	7 (16)	10 (9)	0.19
Days of symptoms until admission	3 (1–5)	8 (6–10)	<0.01
Days from symptoms initiation to IFN	5 (1–6)	10 (8–11)	<0.01
Fever at admission	39 (85)	73 (68)	0.034
Low SpO_2_ < 94% at admission	36 (78)	64 (60)	<0.01
eGFR (MDRD)	67 (52.2–82)	75.4 (67.1–90.2)	0.122
eGFR < 60 mL/min/1.73 m^2^	18 (39)	26 (24)	0.086
Lymphocyte count < 900/μL	25 (55)	60 (56)	0.854
NLR	5.84 (3–12.6)	6.42 (3–13.2)	0.867
Platelets < 150,000/μL	13 (29)	21 (20)	0.197
INR ≥ 1.3	14 (30)	14 (13)	<0.001
D-dimer levels (ng/mL)	818 (606–1408)	750 (426–2347)	0.839
Lactate dehydrogenase ≥ 250 U/L	37 (80)	89 (83)	0.838
C-reactive protein (mg/L)	311 (101–455)	146 (52–380)	<0.01
Procalcitonin	0.12 (0.06–0.25)	0.08 (0.03–0.9)	0.894
Corticosteroids	38 (83)	94 (88)	0.765
Time on CE therapy (days)	10 (4–16)	13.5 (7–22)	0.024
ICU	12 (26)	34 (32)	0.482
Days from symptoms initiation to ICU	9 (6–13)	17 (11–23)	<0.01
Days from admission to ICU	4.5 (1–12)	7 (2–14)	0.447
SAPS II at ICU admission	37 (28–51)	32 (26–40)	0.288
Apache II score at ICU admission	16 (10–33)	13 (10–18)	0.224
Mortality at day 30	23 (50)	37 (35)	0.242
Composed outcome (ICU/Death)	29 (63)	59 (55)	0.364

Abbreviations: EIT, early interferon-β treatment; LIT, late interferon-β treatment; No IT, no interferon-β treatment; BMI, body mass index; eGFR, estimated glomerular filtration rate; SpO_2_, peripheral capillary oxygen saturation; NLR, neutrophil-to-lymphocyte ratio; INR, international normalized ratio; CE, corticosteroids; SAPS II, Simplified Acute Physiologic Score II; APACHE II, Acute Physiology and Chronic Health Disease Classification System II; ICU, intensive care unit.

**Table 3 jcm-11-06952-t003:** Cox regression analysis of factors associated with the time to composed variable ICU admission/death. The variable IFN was no longer associated after adjusting with age, sex, and severity. Data are presented as median, interquartile range (IQR), No. (%). *p*-values are calculated by *χ*^2^, Fisher’s test, or Mann–Whitney’s *U* test.

Variable	IT (*n* = 42)	No IT (*n* = 42)	*p*-Value
Female sex	15 (36)	16 (39)	0.783
Age (range)	62 (21–87)	62 (21–87)	0.993
BMI (Kg/m^2^)	29.8 (24.9–34.1)	28.4 (26–30.1)	0.347
Hypertension	24 (50)	32 (75)	0.988
Diabetes	9 (21)	8 (18)	0.877
Obesity (BMI > 30)	16 (47)	9 (22)	0.389
Chronic kidney disease (eGFR < 60 mL/min/1.73 m^2^)	4 (10)	3 (7)	0.19
Days of symptoms until admission	7 (5–9)	9 (6–15)	0.263
Days from symptoms initiation to IFN	9 (6–11)	-	-
Fever at admission	32 (75)	29 (68)	0.765
Low SpO_2_ < 94% at admission	18 (43)	18 (43)	0.987
Serum creatinine	0.81 (0.77–0.96)	0.8 (0.73–0.92)	0.496
eGFR (MDRD)	86.3 (70.2–95.9)	82.2 (78.6–96.6)	0.623
Lymphocyte count (×10^3^/μL)	1.29 (1.03–1.78)	1.58 (1.38–1.91)	0.046
NLR	2.95 (1.3–4.3)	2.91 (2.3–3.4)	0.883
Platelets < 150,000/μL	15 (36)	3 (7)	<0.001
INR ≥ 1.3	2 (5)	15 (35)	0.04
D-dimer levels (ng/mL)	611 (394–926)	316 (280–764)	0.136
Lactate dehydrogenase ≥ 250 U/L	32 (75)	29 (70)	0.755
C-reactive protein (mg/L)	108 (77–212)	105 (55–124)	0.698
Procalcitonin	0.08 (0.01–1.3)	0.07 (0.01–0.8)	0.877
Corticosteroids	32 (75)	32 (75)	0.91
Time on CE therapy (days)	11 (7–14)	21 (9–24)	0.042
ICU	9 (21)	15 (36)	0.237
Days from symptoms initiation to ICU	17 (10–22)	16 (12–21)	0.137
Days from admission to ICU	8 (3–14)	6 (3–8)	0.837
SAPS II at ICU admission	35 (26–40)	36 (30–49)	0.041
Apache II score at ICU admission	13 (10–18)	14 (11–21)	0.143
Mortality at day 30	15 (36)	13 (32)	0.665
Composed outcome (ICU/Death)	16 (39)	16 (39)	0.99

Abbreviations: IT, interferon-β treatment; No IT, no interferon-β treatment; BMI, body mass index; eGFR, estimated glomerular filtration rate; SpO_2_, peripheral capillary oxygen saturation; NLR, neutrophil-to-lymphocyte ratio; INR, international normalized ratio; CE, corticosteroids; SAPS II, Simplified Acute Physiologic Score II; APACHE II, Acute Physiology and Chronic Health Disease Classification System II; ICU, intensive care unit.

**Table 4 jcm-11-06952-t004:** Comparison of matched patients according to propensity score.

Variable	Hazard Ratio	95% Confidence Interval	*p*-Value
Age (years)	1.015	1.006–1.024	0.001
Lymphocyte count < 900 cells/mm	1.576	1.225–2.028	<0.001
C-reactive protein ≥ 75 mg/L	1.694	1.217–2.359	0.002
eGFR < 60 mL/min/m	1.55	1.205–1.995	0.001
Use of corticosteroids	0.929	0.915–0.943	<0.001
NLR	1.012	1.002–1.21	0.02
IFN therapy	1.149	0.794–1.662	0.461

eGFR, estimated glomerular filtration rate (MDRD equation), IFN, use of sc interferon, NLR, neutrophile-to-lymphocyte ratio. Other variables not significant in the multivariate analysis: sex (*p =* 0.157), obesity (*p =* 0.082), HTA (*p =* 0.21), platelets < 150 × 10 (*p =* 0.478).

## Data Availability

Original data are available upon reasonable request.

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
