# Peer review of "Lost in Translation: Evaluation of Subcutaneous Interferon-β Treatment for SARS-CoV-2 Infection in Real Life"

_jcm, 2022, doi:10.3390/jcm11236952_

Round 1
Reviewer 1 Report
Well written and extensively analyzed. The authors report on the experience of using interferon in hospitalized patients and clearly demonstrate the time limited nature of employing antiviral therapy in the of treatment of patients early during the viremic phase of Covid -19 illness. It is pertinent to note that even the early treatment patients actually were not "early treatment " as these patients were already hospitalized. Patients during the first wave of the pandemic in my honest opinion were in the pulmonary phase of the disease and as one would predict administration of this agent would lead to poorer outcomes. The author should mention this about what they classify as early treatment
Author Response
Well written and extensively analyzed. The authors report on the experience of using interferon in hospitalized patients and clearly demonstrate the time limited nature of employing antiviral therapy in the of treatment of patients early during the viremic phase of Covid -19 illness. It is pertinent to note that even the early treatment patients actually were not "early treatment " as these patients were already hospitalized. Patients during the first wave of the pandemic in my honest opinion were in the pulmonary phase of the disease and as one would predict administration of this agent would lead to poorer outcomes. The author should mention this about what they classify as early treatment
---- Reply: We agree with the reviewer, as the division of IFN therapy as early or late could be confusing. We defined early IFN for those patients with less of 7 days of clinical evolution, for comparison with similar studies, despite severe presentation and hospitalization. We acknowledge this limitation in the Discussion section.
Reviewer 2 Report
This study is a retrospective evaluation of the effectiveness of interferon-B treatment in SARS-CoV-2. I think this study is meaningful because it can be used as evidence to show that IFN-b is not recommended for this reason even in the reality that IFN-b is not strongly recommended in the guidelines for the treatment of SARS-COV-2.
Most of this study is well written, but I think it can be a better manuscript if several points are supplemented.
Methods)
1. In this study, it is necessary to describe the specific definition of ICU/death. If possible, in addition, systematic description of the defined outcomes in this study is required.
Results)
1. In Table 1, IT is divided into EIT and LIT. The table's LIT is written as LI. In addition to IT and non-IT groups, it would be better if you could present data comparing EIT and LIT.
2. Multiple items of variable in Table 1 overlap. It is necessary to simplify the variables including renal function, etc..
3. The result of tocilizumab is in the manuscript, but it is not specified in the table 1.
4. Many variables in Table 1 are different between IT and non-IT groups. The large number of statistically different variables is because there are too many differences between the numbers of the IT group and the control group. Therefore, it is necessary to classify the variables that have a truly meaningful difference among these variables, and to describe their interpretation in the discussion. It is necessary to confirm the opinion of a statistician in this part.
5. In the sentence just above Figure 1, it is described that the outcome is different, but it is necessary to describe what kind of outcome the difference is.
6. In Figure 1, the vertical axis is ICU admission/death, but it needs to be specified accurately. In addition to the legend, it is better to indicate the group according to the color separately.
7. In Table 2, HR should be expressed as adjust HR.
8. Depending on the definition of ICU/mortality in Table2, Cox regression may not be possible. We would appreciate it if you could inform us about the statistical method even if it is not included in the text.
9. In table 2, it would be better to show the results of steroids and tocilizumab in addition to IFN therapy in the variables.
10. It is necessary to describe the interpretation of this research team in the discussion regarding the variables that showed statistically significant results in Table 3.
Author Response
This study is a retrospective evaluation of the effectiveness of interferon-B treatment in SARS-CoV-2. I think this study is meaningful because it can be used as evidence to show that IFN-b is not recommended for this reason even in the reality that IFN-b is not strongly recommended in the guidelines for the treatment of SARS-COV-2.
Most of this study is well written, but I think it can be a better manuscript if several points are supplemented.
Methods)
- In this study, it is necessary to describe the specific definition of ICU/death. If possible, in addition, systematic description of the defined outcomes in this study is required.
Reply: We used this composite definition because not all patients had access to ICU during the first wave of COVID-19 (older, comorbidities, severe presentation...) biasing the response to IFN use (indeed, IFN use could be an “acceptable” alternative for those with lower ICU admission criteria or low probability of response). On the other hand, a lot of patients met ICU criteria and they were admitted to the ICU with similar presentation to IFN treated patients, changing the probability of death during admission. Thus, a composite definition avoid bias. We presented separately the rate of ICU and death (Table 1). We include these differences in the Methods (3rd paragraph) and Discussion section.
Results)
- In Table 1, IT is divided into EIT and LIT. The table's LIT is written as LI. In addition to IT and non-IT groups, it would be better if you could present data comparing EIT and LIT.
Reply: We separately showed a comparison between IT and no IT (Table 1) and we add now a comparison between EIT and LIT (Table 2), and a short commentary about these results.
- Multiple items of variable in Table 1 overlap. It is necessary to simplify the variables including renal function, etc..
Reply: We simplify some variables in the Table 1 and 2(serum creatinine) but we thing that the remaining variables could help to the readers in having a wide idea of included patients.
- The result of tocilizumab is in the manuscript, but it is not specified in the table 1.
Reply: We apologize. Initially, we tried to analyze the use of tocilizumab, but after an in depth revision we decided to exclude related data about this drug (it was used after ICU admission in some cases, even repeated doses, other in those patients not receiving IFN,..). Thus, we decided that we did not have consistent data for analyzing the use of tocilizumab.
- Many variables in Table 1 are different between IT and non-IT groups. The large number of statistically different variables is because there are too many differences between the numbers of the IT group and the control group. Therefore, it is necessary to classify the variables that have a truly meaningful difference among these variables, and to describe their interpretation in the discussion. It is necessary to confirm the opinion of a statistician in this part.
Reply: The IT and no IT groups are completely different in most of variables, as highlighted by the reviewer. IFN group reflected those patients with a bad situation and few therapeutic options, whereas no IFN group is a mixed of patients in different clinical situations. After consultation with our statisticians (JZR, see at the COVID-19 irycis team), they recommended us to evaluate those nearly significant variables (p<0.1) that we consider clinically important for the understanding of the work, avoiding those variables that lead to interaction. The multivariate analysis was then repeated avoiding redundant variables.
- In the sentence just above Figure 1, it is described that the outcome is different, but it is necessary to describe what kind of outcome the difference is.
Reply: We initially included in the Figure 1 the time to ICU/death (composed variable), but this figure has been removed after suggestion by the editor (figure leading to unclear message)
- In Figure 1, the vertical axis is ICU admission/death, but it needs to be specified accurately. In addition to the legend, it is better to indicate the group according to the color separately.
Reply: As mentioned in the former paragraph, the Figure 1 has been deleted. Thanks for your comment
- In Table 2, HR should be expressed as adjust HR.
Reply: We agree. Thanks for the correction.
- Depending on the definition of ICU/mortality in Table2, Cox regression may not be possible. We would appreciate it if you could inform us about the statistical method even if it is not included in the text.
Reply: Cox regression was used to analyze the impact of IT on time to ICU/mortality during admission. Variables of clinical interest with a P value <0.10 in univariate comparisons and those considered of clinical importance were entered into the multivariate models. The variables in the models were selected manually using a backward stepwise process. Interactions and collinearity were evaluated. Sensitivity analyses were performed. It is included in the Methods section.
- In table 2, it would be better to show the results of steroids and tocilizumab in addition to IFN therapy in the variables.
Reply: We agree and thanks for the comment. After a revision of our statistical analysis, we now add the variable “use of CE” in the multivariate analysis, and it was found now as statistically significant (and as mentioned we did not have sufficient data on tocilizumab for a correct analysis).
- It is necessary to describe the interpretation of this research team in the discussion regarding the variables that showed statistically significant results in Table 3.
Reply: In this brief report, we focused our work was in the role of IFN, since it was the main objective of this article.
Reviewer 3 Report
Τhis is an extremely timely and original clinical study that evaluates the therapeutic effect of IFN-β treatment in a large cohort of patients hospitalized with COVID-19 during March-April 2020 with clear results and conclusions that especially with regard to the progression of the disease, are reporting the restriction of its beneficial therapeutic effect mainly in severe cases.
Author Response
Τhis is an extremely timely and original clinical study that evaluates the therapeutic effect of IFN-β treatment in a large cohort of patients hospitalized with COVID-19 during March-April 2020 with clear results and conclusions that especially with regard to the progression of the disease, are reporting the restriction of its beneficial therapeutic effect mainly in severe cases.
Reply: Thanks. We highlight the lack of beneficial effect in the Discussion section, as observed by the reviewer (2ndand 3rd paragraph).